# The Role of Magnetic Transcranial Stimulation in the Diagnosis and Post-Surgical Follow-Up of Cervical Spondylotic Myelopathy

**DOI:** 10.3390/ijerph20043690

**Published:** 2023-02-19

**Authors:** Fernando Vázquez-Sánchez, María del Carmen Lloria-Gil, Ana Isabel Gómez-Menéndez, Francisco Isidro-Mesas, Ana Echavarría-Íñiguez, Javier Martín-Alonso, Jerónimo González-Bernal, Josefa González-Santos, Anna Berardi, Marco Tofani, Giovanni Galeoto, Beatriz García-López

**Affiliations:** 1Neurophysiology Department, Burgos University Hospital, 09006 Burgos, Spain; 2Neurology Department, Valladolid Clinical University Hospital, 47003 Valladolid, Spain; 3Neurosurgery Department, Burgos University Hospital, 09006 Burgos, Spain; 4Department of Health Sciences, Burgos University, 09001 Burgos, Spain; 5Department of Human Neurosciences, Sapienza University of Rome, 00185 Rome, Italy

**Keywords:** myelopathy, spinal cord, cervical spondylotic myelopathy

## Abstract

Degenerative cervical myelopathy (DCM) consists of spinal cord damage due to its compression through the cervical spine. The leading cause is degenerative. The diagnosis is clinical, and the therapeutic approach is usually surgical. Confirmation of the diagnostic suspicion is done by magnetic resonance imaging (MRI); however, this test lacks functional information of the spinal cord, the abnormality of which may precede involvement in neuroimaging. Neurophysiological examination using somatosensory evoked potentials (SSEPs) and transcranial magnetic stimulation (TMS) allows for an evaluation of spinal cord function, and provides information in the diagnostic process. Its role in the post-surgical follow-up of patients undergoing decompressive surgery is being studied. We present a retrospective study of 24 patients with DCM and surgical decompression who underwent neurophysiological tests (TMS and SSEP) before, 6, and 12 months after surgery. The result of the TMS and the SSEP in the post-operative follow-up did not correlate with the clinical outcome, either subjective or measured by clinical scales at six months. We only found post-surgical improvement of central conduction times (CMCTs) in patients with severe pre-surgical motor impairment on TMS. In patients with normal pre-surgical CMCT, we found a transient worsening with return to baseline at the one-year follow-up. Most patients presented pre-surgical increased P40 latency at diagnosis. CMCT and SSEP were more related to clinical outcomes one year after the surgical procedure and were very useful in diagnosing.

## 1. Introduction

The term degenerative cervical myelopathy [1] (DCM) has recently been proposed to encompass all the pathophysiological compressive mechanisms, the result of which is spinal cord dysfunction at the cervical level. DCM is caused by osteoarticular changes that include spondylosis and disc herniation, ligamentous hypertrophy, as well as its calcification, ossification, and facet arthropathy [2]. The classically used term, still widely used, is cervical spondylotic myelopathy (CSM), which is more restrictive as it does not include all the previously mentioned and potentially harmful mechanisms.

Spinal cord compression by cervical structures triggers a cascade of phenomena that include ischemia, rupture of the blood-spinal cord barrier, and apoptosis [3], causing demyelination and secondary gliosis leading to its dysfunction.

The clinical manifestations of DCM are variable. Patients may be asymptomatic, have isolated cervical pain, radicular-type pain in the upper limbs, or have symptoms secondary to spinal cord compression. These include a pyramidal syndrome, sensory manifestations in the limbs, gait impairment, and loss of sphincter control [4]. These patients may also develop an acute centromedullary syndrome due to cervical hyperextension in a previously spondylotic cervical spine [5].

The initial diagnosis is clinical, based on the anamnesis, and supported by an exhaustive neurological examination. Clinical data also allows a later re-evaluation of the progression in the conservative long-term follow-up or in post-surgical evolution. There are several scales to assess the clinical injury degree. The modified Japanese Orthopedic Association (mJOA) score [6,7] is one of the most widely-used despite the fact that interobserver variability may be significant [8]; Table A1 provides complementary material. The Nurick score [9,10], more focused in gait impairment, allows for the classification of the degree of affection in six grades; Table A2 provides complementary material. There are many other scales. Nevertheless, despite these scale’s limitations, each’s overall usefulness seems comparable [11]. It should be noted that the functional disorder may be present before the lesion is visible on an MRI. In this way, clinical evaluation remains essential and neurophysiological study may be helpful after the first clinical evaluation.

Confirmation of spinal cord damage is performed by an MRI, with common findings such as the decreased diameter of the spinal canal, hypo intensity of the spinal cord on T1-weighted images, hyperintensity on T2-weighted images, loss of cerebrospinal fluid image, and spinal deformation [4,12]. The main limitation of neuroimaging is the lack of functional information of the spinal tracts. Nevertheless, it predicts postsurgical outcome [13,14]. Neurophysiological evaluation through somatosensory evoked potentials (SSEPs) and magnetic transcranial stimulation (TMS) can detect the functional alteration of the spinal cord at any time during the natural evolution of the disease [15,16]. Currently, both SSEPs and TMS provide information, not only diagnostic but also prognostic. Several studies have assessed prognostic information from a functional point of view through the correlation of the central conduction time obtained by TMS with the clinical evolution of the patients [10,16,17].

SSEP is a functional evaluation method of the posterior medullary cords whose alteration appears early in the CSM, especially in the lower extremities. They have been used in the initial assessment and later in the follow-up to evaluate post-surgical changes. It is a robust, simple test, but can also be time-consuming.

TMS allows an evaluation of the integrity of the pyramidal pathway by magnetic stimulation of the primary motor cortex and subsequent measurement of central motor conduction times (CMCTs) [18]. CMCT alterations can be due to desynchronization, temporal dispersion, conduction block, or axonal degeneration in the fastest conducting fibers [19]. The correlation of TMS with clinical and MRI findings has been previously studied [20,21]. Some of these studies have shown that the correlation between changes in post-surgical TMS and objective clinical parameters measured using clinical scores is more accurate than the subjective parameters provided by patients. They also found that pre-surgical prolonged CMCT had poor prognostic value for predicting recovery [22].

According to one study, TMS abnormality with prolonged CMCTs correlated with the clinical outcome better than MRI findings. They defined MRI abnormalities as canal stenosis without visible signs of myelopathy on T2 hyperintensity or short-time investment recovery (STIR) [17].

The therapeutic approach of CSM can be surgical or conservative, depending on the patient. Conservative treatment is usually considered in case of minor symptoms, cervical stenosis, and absence of visible signs of myelopathy on an MRI. Surgical treatment prevents progression and allows, in less severe cases of compression, a better functional recovery, as well as an improvement in the quality of life of patients. Therefore, surgical treatment is increasingly recommended [4]. Various studies have shown that the most significant improvement is achieved after surgery in cases with mild-to-moderate functional involvement evaluated with the different functionality scales, compared to those with severe involvement where part of the damage could be irreversible [15]. Good postsurgical outcome is related to younger age, absence of radiographic myelopathy signs, less duration of symptoms before surgery and the presence of sensory instead of motor or gait disturbance as first symptoms [23,24,25]. The aim of our study was to correlate the clinical outcome measured by scales with the neurophysiological sensory and motor tests evolution.

## 2. Materials and Methods

### 2.1. Study Population

Our population was comprised of 24 patients, recruited between 2018 and 2020, who presented symptoms compatible with CSM (patients whose MRIs showed spondyloarthrosis and a confirmed spinal cord injury understood as an alteration of the intramedullary signal in any of the sequences (T1, T2, STIR)), or as altered functional tests (EMTC or PESS) if the intramedullary signal was normal (n = 1). All the patients had a decompressive surgery indication, according to the surgical team criteria in our hospital, with intraoperative neurophysiological monitoring. All patients underwent a neurophysiological study before surgery (t_0_), and again at 6 (t_1_) and 12 months (t_2_) after surgery. We included patients either with anterior or posterior cervical surgical approaches. The exclusion criteria were patients under 18 years of age, patients who did not want to participate in the follow-up due to a lack of tolerance of the diagnostic test (n = 2), and those in which their previous pathology (central or peripheral cause) avoided obtaining a reliable signal on the neurophysiological tests (polyneuropathies, other causes of myelopathy, spinal cord structural injuries such as syringomyelia, or acute traumatic causes (n = 4)) (Figure 1).

The patients voluntarily decided to participate in the follow-up study. They signed their informed consent for participation and the Burgos and Soria Hospital Ethics Committee validated the study with the registration number CEIC 1618.

### 2.2. Neurophysiological Study

The tests was comprised of SSEP as a method of sensory pathway exploration, and TMS to study the motor pathway. We used 32-channel amplifier Natus equipment. SSEPs were performed with stimulation of both median (wrist) and posterior tibial nerves (internal malleolus), bilaterally. Technically, we followed the recommendations of the International Federation of Clinical Neurophysiology (IFCN) [26], with peripheral recording on Erb point and the popliteal fossa, as well as lumbar, cervical, and cortical locations. The cortical latencies of the N20 and P40 waves were assessed. The cortical latency of the N20 and P40 waves were assessed with normal range values according to Delisa and Chiappa for upper and lower limbs, respectively [27,28]. For the statisctical correlation study we searched for variations of latency of the cortical N20 and P40 evoked potentials.

TMS was performed with a single-stimulus Magstim stimulator, with a 14 cm diameter circular coil, stimulating the primary motor cortex with recording in the *abductor pollicis brevis* muscles in the upper limbs and *tibialis anterior* in the lower limbs, bilaterally. We used a single positive pulse of a 0.02 ms duration both in cortex and in cervical and lumbar stimulation, at supramaximal intensity in upper limbs, and 100% intensity in the motor cortex for the loser limbs. We assessed CMCT with cervical and lumbar stimulation, with reference values according to Abbruzzese and Barker [18,29].

### 2.3. Cervical MRI

All patients underwent a cervical 1.5 Tesla MRI at t_0_. Abnormalities that defined cervical myelopathy in MRIs consisted of a hyper signal on T2, short time investment recovery (STIR), or hypo intensity on T1 sequences.

### 2.4. Study Design

We carried out an observational, descriptive, retrospective study of patients surgically treated for cervical canal decompression between 2018 and 2020. The patients underwent a pre- and post-surgical neurophysiological follow-up study 6 and 12 months after surgery. We retrospectively collected the clinical data of the patients for whom the Nurick and mJOA scales had been completed in consultation at diagnosis and six months after surgical decompression.

The main objective was to correlate neurophysiological tests (SSEP and TMS) to the patient’s subjective clinical improvement and the Nurick and mJOA score variations.

### 2.5. Subjective and Objective Clinical Measures

We used the patient’s clinical symptoms reported as subjective measures at baseline, and recovery impression at six months (improvement versus no improvement). We considered objective measures, clinical signs explored by neurological examination (sensory deficits, motor deficits, and gait impairment attending to the gait pattern and tandem walking). A classification of the degree of myelopathy was also carried out according to the Nurick and mJOA scales, using the defined cut-off scores established by Tetreault and colleagues for the last scale: mild myelopathy between 15 and 17 points, moderate between 12 and 14 points, and severe below 11 points [30].

As objective measures, we observed changes in the neurological examination and the neurophysiological studies, specifically in CMCT and SSEPs latencies.

### 2.6. Statistical Analysis

An anonymized database was created for later statistical analysis with SPSS software (Version 28). The subjective clinical evolution of the patients six months after surgery was studied using the dichotomous variable improvement versus no improvement, and we compared it with the Nurick and mJOA clinical score changes at t_1_. We performed a Student’s t-test (1-tailed distribution; paired) to assess the degree of affectation in the pre-and post-surgical Nurick scale. We also performed this test (1-tailed distribution; paired) to assess the pre- and post-surgical mJOA scale’s improvement. The changes in the mJOA scale were evaluated according to the degree of involvement at diagnosis (mild, moderate, or severe). Sensory symptoms, motor, and gait impairment, and neurological examination clinical changes were evaluated in detail. We also calculated the minimum clinically important difference (MCID) of the mJOA according to the distribution-based methods [31]. Finally, we studied the modifications in SSEPs latencies and CMCT, considering each limb individually at t_0_, t_1_, and t_2_.

## 3. Results

### 3.1. Characteristics of the Population at Baseline (t_0_)

A total of 24 patients were included in our study; 66% (n = 16) were men and 34% women (n = 8). The mean age was 56.6 (with a median of 55 years). At t_0_, the mean Nurick score was 2.37 points, and 13 points on the mJOA scale. On this last scale and according to the Tetreault and colleagues criteria, 29.1% of the patients (n = 7) presented a mild pre-surgical degree of myelopathy, 41.6% (n = 10) a moderate degree, and 29.1% (n = 7) a severe degree. Regarding symptoms, 92% of the patients (n = 22) presented some sensory disorder, 75% (n = 18) had a motor disturbance, and 71% of the patients (n = 17) had gait impairment (subjective perception of gait impairment included here). Finally, 83% of the patients (n = 20) presented with some abnormality in the neurological examination (including objective gait impairment). Up to 85% of the patients in the mild group presented sensory symptoms. Based on neuroimaging, 96.4% (n = 22) of the patients had clear signs of myelopathy on MRIs. Only 4.2% (n = 1) did not present myelopathy lesions in MRIs. Up to 58% had myelopathy at a single spinal cord level, 20.7% had myelopathy at two levels, and 13% had it at three levels. The most affected level was C5–C6.

Concerning motor conductions, 65% of CMCT were pathological in the upper limbs and 69.1% in the lower limbs at t_0_.

Regarding sensory spinal conduction at diagnosis, 49.4% of the examinations presented pathological values, with marked differences in the distribution of abnormalities between upper and lower limbs. Only 27% had altered SSEPs in the upper limbs (N20 cortical latency) while 79% had a prolonged SEEP in the lower limbs (P40 cortical latency). Table 1 shows CMCT and SSEP at t_0_ and t_2_ subdivided by groups according to mJOA grades.

### 3.2. Post-Surgical Subjective Clinical Situation

The patients reported the subjective clinical change through a clinical interview at t_1_. A total of 75% (n = 18) reported subjective post-surgical improvement in any clinical aspect (pain, paresthesia, dexterity, gait, sphincter control), and the remaining 25% (n = 6) reported no improvement. Of this last group, two patients did not present changes in the mJOA scale. One patient lost one point, another gained one point, and another gained two points. Finally, one patient did not recognize improvement despite gaining 4 points on the scale (the gait went from aid needed to walk at t_0_ to an insecure but autonomous gait at t_1_). In the group of patients that reported improvement, three patients had gained one point, three patients two points, eight patients had gained three points, two patients had gained four points and, finally, one patient had gained five points. The MCID of the mJOA calculated with the distribution-based methods was 1.1.

### 3.3. Clinical Evaluation and Scales Scores at t_0_ and t_1_

Concerning the clinical improvement assessed by anamnesis and clinical exams, sensory disturbance improved in 58% of the patients, motor impairment in 66%, gait in 58%, and any aspect of the neurological examination in 96%.

The mean score on the mJOA scale was 13 at t_0_ with a value of 15.6 at t_1_. This change means a global improvement of 2.6 points. Clinical post-surgical changes measured by mJOA were statistically significant (*p* < 0.01 = 5.266 × 10^−8^).

Divided by the severity, and according to Treteault and colleagues [30], the mean pre-surgical mJOA in the severely affected group at t_0_ was 10, reaching a mean of 12.8 at t_1_ (an increase of 2.8 points). In the moderate group, the mean was 13.3 at t_0_, reaching 16.5 at t_1_ (an increase of 3.2 points). Finally, in those with mild involvement, the mean score was 15.7 at t_0_ and 16.57 at t_1_ (an increase of 0.87 points).

The mean score on the Nurick scale was 2.37 (median 3) at t_0_ and 1.25 at t_1_ (a mean improvement of 1.12 points) after surgery. In the Nurick scale, the clinical changes between t_0_ and t_1_ were statistically significant (*p* < 0.01 = 4.066 × 10^−5^) (Table 2 and Table 3).

### 3.4. Evolution of CMCT and SSEP at t_0_ and t_1_

From all patients, the percentage of TMS and SSEP abnormalities at t_0_ was not equally distributed in the upper and lower limbs. Regarding motor impairment, CMCT in any of the upper limbs was abnormal in 75% of patients, while in the lower limbs it was in 87.5%. Concerning sensory impairment, 29.2% had prolonged N20 latency in at least one of the upper limbs, and 87.5% presented prolonged P40 latency in at least one of the lower limbs. All patients with prolonged N20 had prolonged P40. In the TMS, we observed that 70.8% of patients presented improvement in the CMCT in more than one standard deviation in any limbs at t_2_ compared to t_0_. SSEP latencies improvement with the same criteria was observed in 45.8% of the patients. Table 1 presents this data subdivided by the mJOA score at t_0_.

There were no significant differences in CMCT at t_1_ compared to CMCT at t_0_ in patients stratified by the degree of mJOA affectation at t_0_ with the univariate analysis of variance. A marked trend towards improvement in central conduction time was only observed in the subgroup of patients whose CMCT at t_0_ showed a marked alteration in CMCT, with values above 140% of the normal limit value (12 milliseconds in EESS and 23.43 in lower limbs) compared to those who preoperatively had a TCC within normal limits (*p* = 0.069 in univariate analysis of variance). In this last group of patients with normal CMCT at t_0_, CMCT notably increased its value (16%) at t_1_ and returned to baseline at t_2_ with a minimal difference (2%).

Concerning the sensory spinal cord conduction study, an improvement trend in SSEP was observed only at t_2_ according to the paired samples test (*p* = 0.068). There were no differences in the values in the univariate analysis of variance by stratifying the sample according to the Treteault clinical grades by mJOA at t_0_.

Among patients with subjective reported improvement, up to 28,6% presented measured changes in both motor and sensory neurophysiological tests. Isolated CMCT improvement was observed in 35.7%, isolated SSEP improvement in 14.3%, and no changes in the neurophysiological tests in 21.4%. If we consider only the motor study in these patients independently from sensory function, 64.3% presented improvement. Sensory improvement regardless of motor function was found in 42.8% of the patients included in this subgroup. From patients who did not report changes for the better (41.7%), we observed both motor and sensory neurophysiological improvement in 40%, only motor in 40%, and only sensory improvement in 10%. There was no change in 10%.

## 4. Discussion

We present a series of 24 patients. The mean age of our patients was 56.6 years which is not far from that found in other studies [15,17,21,32].

The evaluation of the patients according to their subjective perception of outcome, using the dichotomy of improvement versus no improvement, is not far from the mJOA scale assessed evolution at t_1_. While 75% of patients reported amelioration at t_1_, the mJOA scale showed an improvement in 91.6% after surgery at the same time. All patients except two reporting improvement had shown an increase of at least two points in the mJOA score. The calculated MCID was 1.1, but few patients recognized changes of a single point on the mJOA scale. When scales changed by two points, they reported either improvement or clinical stability. In increments greater than two points, they referred clear improvement. A single patient reported no improvement despite an increase of four points on the scale, which may be due to unrealistic expectations regarding surgery, considering that she reached the ability to walk unaided, having the need of a walker at baseline according to her medical history record. Zhou and colleagues did not find any correlation at 3 months between subjective improvement and mJOA changes after surgery among 129 patients [33].

Clinical scales performed after surgery showed global improvement. In the mJOA scale, we found 21% of patients in a mild degree at baseline, increasing up to 75% at t_1._ We observed similar changes in the Nurick scale; globally, more severe grades 3 and 4 reduced, and the percentage of patients in grades 0, 1, and 2 increased. At t_0_, 8% of patients had a grade of 0, counting a 25% at t_1_. Those in grade 1 went from 21% to 42% of all patients. Regarding the evolution of neurophysiological tests compared to scales, we found no correlation. Moreover, we observed a transient CMCT worsening at t_1_ in patients with normal CMCT values at t_0_, which returned to baseline values at t_2_. Nakanishi and colleagues found a correlation between the mJOA score and CMCT at the one-year follow-up [32]. Capone and colleagues found improvement only in lower limbs and in the moderate or mild impairment group [15]. We only found this correlation in patients with severe motor conduction impairment, with CMCT at t_0_ > 140% of normal limits with no difference between upper or lower limbs. Although we lack data of mJOA scores at one year, the clinical improvement had already occurred at the 6-month follow-up. Lo and colleagues found improvement in cases with upper TMS abnormalities. They suggest that upper limb CMCT abnormality reflects a more severe affectation of the corticospinal tracts, as the upper limbs are placed more medially in the cervical spinal cord [21]. They also found that upper limb CMCT alteration at diagnosis is an independent predictor of good surgical outcome. These results support our own results. 71.4% of patients in the severe degree of mJOA improved, indicating that, even in cases with severe involvement from a neurophysiological point of view, surgery can be beneficial not only to prevent long-term worsening but also to recover some of the lost functionality. In our patients with a moderate grade, clinical improvement occurred independently of changes in neurophysiological tests. This fact may suggest that follow-up through neurophysiological tests is less valuable than clinical scale measurements, at least in this group and, if done, it should be performed at least one year after surgery, which seems to correspond to the outcome in the rest of the groups of impairment. As opposed to our results, Capone and colleagues described an improvement in CMCTs in the lower limbs in those patients whose pre-surgical clinical involvement was mild to moderate, compared to an absence of changes in patients whose initial clinical severity was more significant [15]; Jaskolski measured pre- and post-surgical CMCTs in patients with cervical spondylosis without finding valuable predictive information for clinical outcomes [34].

Concerning SSEP, in some patients, there was a significant improvement in its values at t_2_, with no differences between groups classified by the mJOA scale. We observed any degree of improvement in any of the four limbs in 42.9% in the mild impairment group, 50% in the moderate impairment group, and 42.3% in the severe group, a similar pattern to the motor changes observed by TMS. Narbone et al. have found SSEP to help in the assessment of disease severity and for monitoring and quantifying function in the course of patient recovery [17].

All patients except one had an MRI hyper-signal on T2 (STIR) before surgery. This alteration means gliosis and, therefore, a definitive lesion [35]. The appearance of a low-intensity signal on T1 would be more related to a worse post-surgical prognosis, although hyperintensity on T2 may be a good marker of poor prognosis in the absence of hypointensity on T1. A DWI sequence detects spinal cord damage before T1 and T2 images are pathological so that it can detect spinal cord dysfunction earlier, with the potential benefit of early detection [36,37]. However, on many occasions, the disease diagnosis is made at this stage due to multiple causes such as late consultation of the symptoms or initial confusion with other pathologies such as carpal tunnel syndrome. In our inclusion process, we did not find patients without myelopathy signs visible on MRIs, except for the one included. We can conclude of a late intervention concerning the onset of symptoms. Until not long ago, the general trend was to use hyper-signal on MRI as one of the criteria for the surgical approach. In a two-year follow-up study, Deftereos and colleagues found no clinical deterioration in patients with cervical canal stenosis when the spinal cord function was preserved (assessed by neurophysiological tests and normal MRI cervical spinal cord signals). Addotionally, they reported that TMS is a better long-term clinical outcome predictor compared to MRI [17].

Considering the high percentage of patients with abnormal neurophysiological tests at diagnosis in the mild impairment mJOA group (up to 71.4% of prolonged CMCT in upper and in the lower limbs and up to 85.7% SSEP abnormalities in the lower limbs), we think that neurophysiological tests are especially helpful supporting the initial clinical diagnosis. In patients with compatible symptoms, with or without abnormalities in the neurological examination, and normal spinal cord signal in MRIs, pathological TMS, and PESS would be enough to assess spinal cord dysfunction and, therefore, to consider the surgical treatment. Some authors have suggested that in patients who are candidates for surgery, the treatment should be considered at an early stage of the disease [15]. Based on these data, we believe it would be interesting to evaluate the inclusion of neurophysiological tests in the therapeutic decision algorithm. For this purpose, further studies are necessary, including a branch of patients with an early-stage diagnosis and surgical approach. This group would be formed of patients with a milder degree of involvement (compatible symptoms, normal MRI, and functional alteration verified by neurophysiological tests) to assess the post-surgical evolution and correlate it with the clinical course of patients. CMCTs performed in more than one muscle per limb could be helpful in detecting an earlier functional spinal cord impairment. The surgical treatment in these patients would invest in avoiding post-surgical neurological sequelae. According to Deftereos and colleagues, abnormality of TMS or SSEP would be mandatory before deciding on a surgical approach helping in the setting of surgical indication [17]. Finally, and considering our results and the difficulty in assessing the correlation of clinical and neurophysiological parameters at follow-up through neurophysiological tests, we should think that clinical follow-up after surgery is enough to assess outcomes. Our study is limited by the small number of patients recruited and the loss of some patients due to neurophysiological test intolerance (n = 2). The inclusion process and the follow-up found difficulty derived from the COVID-19 pandemic since the years analyzed were between 2018 and 2020.

The clinical evaluation was performed only at six months and not at 12 months, to be able to compare it with the new neurophysiological examination at t_2_.

Different observers filled the mJOA scale at t_1_.

The study lacked a control group or a group without MRI spinal cord abnormalities to compare.

## 5. Limitations

Our study is limited by the small number of patients recruited. The inclusion process and the follow-up found difficulties derived from the COVID-19 pandemic since the years analyzed were between 2018 and 2020, and some patients were lost due to different causes already explained. The clinical evaluation was performed only at t_1_, and the mJOA scale at t_1_ was filled by different observers. The mJOA was not available 12 months after surgery to compare it with the second neurophysiological examination. Our study lacks a control group or a group without MRI spinal cord abnormalities to compare.

## 6. Conclusions

According to our results, subjective improvements reported by patients correlated with changes measured by the mJOA scale. Most patients recognize changes from two points.

The post-surgical follow-up of patients through neurophysiological tests (TMS and SSEP) is of little use compared to clinical evaluation through anamnesis and neurological examination and follow-up with scales, specially mJOA. In the case of performing follow/up neurophysiological studies, the examination carried out one year after surgery seems more reliable, avoiding misinterpretation due to the transient worsening observed after surgery (six months after surgery) in some patients. We only observed a correlation between evolution in mJOA with CMCT improvement when this time was severely prolonged (CMCT at t_0_ > 140% of the normal upper limit). Neurophysiological tests would have value in the pre-surgical evaluation of patients with suggestive symptoms of DM without MRI spinal cord nor neurological examination abnormalities.

## Figures and Tables

**Figure 1 ijerph-20-03690-f001:**
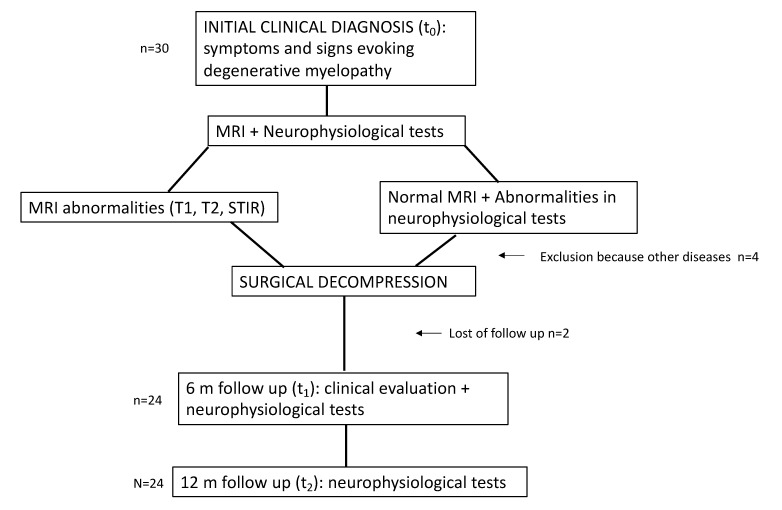
Flowchart of the diagnostic process, surgical treatment, and follow-up.

**Table 1 ijerph-20-03690-t001:** Percentage of patients with abnormal TMS and PESS before surgery and changes after surgery according to mJOA grades in upper and lower limbs.

mJOA at t_0_	mJOA Improvement (≥1 Point)	Abnormal CMCT Upper Limbs (t_0_)	Abnormal CMCT Lower Limbs (t_0_)	Abnormal SSEP Upper Limbs (t_0_)	Abnormal SSEP Lower Limbs (t_0_)	Improvement CMCT Upper Limbs (t_2_)	Improvement CMCT Lower Limbs (t_2_)	Improvement SSEP Upper Limbs (t_2_)	Improvement SSEP Lower Limbs (t_2_)
Severe (29.2%)	85.7%	71.4%	100%	57.1%	100%	50%	71.4%	33.3%	71.4%
Moderate (41.7%)	100%	80%	90%	20%	80%	55.6%	100%	44.4%	50%
Mild (29.2%)	85.7%	71.4%	71.4%	14.3%	85.7%	60%	80%	100%	40%

**Table 2 ijerph-20-03690-t002:** Distribution of patients according to their mJOA mean scores at t_0_ and t_1_.

mJOA Scale	t_0_	t_1_
Mild	29.1% (n = 7)	75% (n = 18)
Moderate	41.5% (n = 10)	16.6% (n = 4)
Severe	29.1% (n = 7)	8.3% (n = 2)

**Table 3 ijerph-20-03690-t003:** Distribution of patients according to their Nurick score classification at t_0_ and t_1_.

NURICK Scale	t_0_	t_1_
Grade 0	8%	25%
Grade 1	21%	42%
Grade 2	17%	21%
Grade 3	33%	8%
Grade 4	21%	4%
Grade 5	0%	0%

## Data Availability

Not application.

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
