# Peer review of "The Role of Magnetic Transcranial Stimulation in the Diagnosis and Post-Surgical Follow-Up of Cervical Spondylotic Myelopathy"

_ijerph, 2023, doi:10.3390/ijerph20043690_

Round 1
Reviewer 1 Report (Previous Reviewer 1)
Dear Authors, thank you very much for implementing the changes required.
Reviewer 2 Report (Previous Reviewer 2)
Retrospective studies are difficult because of inherent unavoidable bias. In your study, the surgical procedures done, the number of different surgeons involved and limitations of data among others. Admittedly the number is small. But the study spurs further efforts necessary to find the true value of SSPE and TMS in surgical treatment of myelopathy resulting from cervical spinal spondylosis. Good effort.
This manuscript is a resubmission of an earlier submission. The following is a list of the peer review reports and author responses from that submission.
Round 1
Reviewer 1 Report
Dear Authors!
Thank you very much for sharing your interesting work. However, there are several major and minor critical points that should be addressed priorly.
Major points:
1. Why did you only include 30 patients and evaluated only 24 of them? Strength statistic is not possible with only 24 patients. Additionally, you sub-divided the cohorts making the groups even smaller and the statistical strength even less powerful. Therefore, authors should include more patients within analysis.
2. Introduction is too long. Authors should focus on DCM describing shortly pathophysiology, symptoms, diagnostics, surgical therapy and its outcome and predictors. Afterwards, authors should present their own work and thoughts on diagnostics and postoperative control.
3. I did not understand Table 1: Analysis of mild/moderate/severe DCM is not presented well. Should you not describe motor deficits of upper and lower extremities, sensory deficits and gait disturbance? How did you analyze gait disturbance? What do you mean with abnormal neurological examination?
4. MCID should be evaluated and analyzed (10.1097/BRS.0000000000001127).
5. Discussion is poorly presented. Authors should focus on their findings (especially neurophysiological changes) and compare it with the literature.
6. Study limitations have to be presented as a single paragraph before the conclusion. Additionally, limitations have to be presented in more detail.
7. Conclusion is too long and should focus on changes in neurophysiological testing. Changes in mJOA are not new.
8. Additionally, important literature is missing:
a. 10.3390/jcm9010062
b. 10.1177/2192568217701914
c. 10.1097/BRS.0000000000001127
d. 10.1097/BRS.0000000000003750
e. 10.1097/BRS.0000000000000678
f. 10.1007/s00586-021-07060-3
g. 10.1038/s41582-019-0303-0
h. 10.1093/neuros/nyy474
i. 10.1093/neuros/nyz178
Minor points:
1. English check has to be performed by a native speaker.
Reviewer 2 Report
Congratulations to the authors for a very well designed, well-done, very relevant study.
Diagnosis and decision to do surgery for cervical myelopathy in cervical spondylosis/stenosis is usually based on clinical diagnosis of the disease, its severity and progression.
The relevance of neurophysiologic tests would be most helpful mainly as the authors have aptly observed in decisions to intervene in mild cervical myelopathy and in predicting and measuring objective response to surgical treatment in severe cases.
Where diagnosis, severity and progression are evident, SSEP, TMS and CCMT may be unnecesssary. More study will be helpful.